# ADVERSARIAL AUDIO SYNTHESIS

**Chris Donahue**
Department of Music
UC San Diego
cdonahue@ucsd.edu

**Julian McAuley**
Department of Computer Science
UC San Diego
jmcauley@eng.ucsd.edu

**Miller Puckette**
Department of Music
UC San Diego
msp@ucsd.edu

## ABSTRACT

Audio signals are sampled at high temporal resolutions, and learning to synthesize audio requires capturing structure across a range of timescales. Generative adversarial networks (GANs) have seen wide success at generating images that are both locally and globally coherent, but they have seen little application to audio generation. In this paper we introduce WaveGAN, a first attempt at applying GANs to unsupervised synthesis of raw-waveform audio. WaveGAN is capable of synthesizing one second slices of audio waveforms with global coherence, suitable for sound effect generation. Our experiments demonstrate that—without labels—WaveGAN learns to produce intelligible words when trained on a small-vocabulary speech dataset, and can also synthesize audio from other domains such as drums, bird vocalizations, and piano. We compare WaveGAN to a method which applies GANs designed for image generation on image-like audio feature representations, finding both approaches to be promising.

## 1 INTRODUCTION

Synthesizing audio for specific domains has many practical applications in creative sound design for music and film. Musicians and Foley artists scour large databases of sound effects to find particular audio recordings suitable for specific scenarios. This strategy is painstaking and may result in a negative outcome if the ideal sound effect does not exist in the library. A better approach might allow a sound artist to explore a compact *latent space* of audio, taking broad steps to find the types of sounds they are looking for (e.g. footsteps) and making small adjustments to latent variables to fine-tune (e.g. a large boot lands on a gravel path). However, audio signals have high temporal resolution, and strategies that learn such a representation must perform effectively in high dimensions.

Generative Adversarial Networks (GANs) (Goodfellow et al., 2014) are one such unsupervised strategy for mapping low-dimensional latent vectors to high-dimensional data. The potential advantages of GAN-based approaches to audio synthesis are numerous. Firstly, GANs could be useful for data augmentation (Shrivastava et al., 2017) in data-hungry speech recognition systems. Secondly, GANs could enable rapid and straightforward sampling of large amounts of audio. Furthermore, while the usefulness of generating static images with GANs is arguable, there are many applications (e.g. Foley) for which generating sound effects is immediately useful. But despite their increasing fidelity at synthesizing images (Radford et al., 2016; Berthelot et al., 2017; Karras et al., 2018), GANs have yet to be demonstrated capable of synthesizing audio in an unsupervised setting.

A naïve solution for applying image-generating GANs to audio would be to operate them on image-like *spectrograms*, i.e., time-frequency representations of audio. This practice of bootstrapping image recognition algorithms for audio tasks is commonplace in the discriminative setting (Hershey et al., 2017). In the generative setting however, this approach is problematic as the most perceptually-informed spectrograms are non-invertible, and hence cannot be listened to without lossy estimations (Griffin & Lim, 1984) or learned inversion models (Shen et al., 2018).

Recent work (van den Oord et al., 2016; Mehri et al., 2017) has shown that neural networks can be trained with autoregression to operate on *raw audio*. Such approaches are attractive as they dispense with engineered feature representations. However, unlike with GANs, the autoregressive setting results in slow generation as output audio samples must be fed back into the model one at a time.

In this work, we investigate both waveform and spectrogram strategies for generating one-second slices of audio with GANs.[1] For our spectrogram approach (SpecGAN), we first design a spectrogram representation that allows for approximate inversion, and bootstrap the two-dimensional deep convolutional GAN (DCGAN) method (Radford et al., 2016) to operate on these spectrograms. In WaveGAN, our waveform approach, we flatten the DCGAN architecture to operate in one dimension, resulting in a model with the same number of parameters and numerical operations as its two-dimensional analog. With WaveGAN, we provide both a starting point for practical audio synthesis with GANs and a recipe for modifying other image generation methods to operate on waveforms.

We primarily envisage our method being applied to the generation of short sound effects suitable for use in music and film. For example, we trained a WaveGAN on drums, resulting in a procedural drum machine designed to assist electronic musicians (demo `chrisdonahue.com/wavegan`). However, human evaluation for such domain-specific tasks would require expert listeners. Therefore, we also consider a speech benchmark, facilitating straightforward assessment by human annotators. Specifically, we explore a task where success can easily be judged by any English speaker: generating examples of spoken digits "zero" through "nine".

Though our evaluation focuses on a speech generation task, we note that it is *not* our goal to develop a text-to-speech synthesizer. Instead, our investigation concerns whether unsupervised strategies can learn global structure (e.g. words in speech data) implicit in high-dimensional audio signals without conditioning. Our experiments on speech demonstrate that both WaveGAN and SpecGAN can generate spoken digits that are intelligible to humans. On criteria of sound quality and speaker diversity, human judges indicate a preference for the audio generated by WaveGAN compared to that from SpecGAN.

## 2 GAN PRELIMINARIES

GANs learn mappings from low-dimensional latent vectors $z \in \mathcal{Z}$, i.i.d. samples from known prior $P_Z$, to points in the space of natural data $\mathcal{X}$. In their original formulation (Goodfellow et al., 2014), a generator $G : \mathcal{Z} \mapsto \mathcal{X}$ is pitted against a discriminator $D : \mathcal{X} \mapsto [0, 1]$ in a two-player minimax game. $G$ is trained to minimize the following value function, while $D$ is trained to maximize it:

$$V(D, G) = \mathbb{E}_{\boldsymbol{x} \sim P_X}[\log D(\boldsymbol{x})] + \mathbb{E}_{\boldsymbol{z} \sim P_Z}[\log(1 - D(G(\boldsymbol{z})))]. \tag{1}$$

In other words, $D$ is trained to determine if an example is real or fake, and $G$ is trained to fool the discriminator into thinking its output is real. Goodfellow et al. (2014) demonstrate that their proposed training algorithm for Equation 1 equates to minimizing the Jensen-Shannon divergence between $P_X$, the data distribution, and $P_G$, the implicit distribution of the generator when $z \sim P_Z$. In this original formulation, GANs are notoriously difficult to train, and prone to catastrophic failure cases. Instead of Jensen-Shannon divergence, Arjovsky et al. (2017) suggest minimizing the smoother Wasserstein-1 distance between generated and data distributions

$$W(P_X, P_G) = \sup_{\|f\|_L \leq 1} \mathbb{E}_{x \sim P_X}[f(x)] - \mathbb{E}_{x \sim P_G}[f(x)] \tag{2}$$

where $\|f\|_L \leq 1 : \mathcal{X} \mapsto \mathbb{R}$ is the family of functions that are 1-Lipschitz.

To minimize Wasserstein distance, they suggest a GAN training algorithm (WGAN), similar to that of Goodfellow et al. (2014), for the following value function:

$$V_{\text{WGAN}}(D_w, G) = \mathbb{E}_{\boldsymbol{x} \sim P_X}[D_w(\boldsymbol{x})] - \mathbb{E}_{\boldsymbol{z} \sim P_Z}[D_w(G(\boldsymbol{z}))]. \tag{3}$$

With this formulation, $D_w : \mathcal{X} \mapsto \mathbb{R}$ is not trained to identify examples as real or fake, but instead is trained as a function that assists in computing the Wasserstein distance. Arjovsky et al. (2017) suggest weight clipping as a means of enforcing that $D_w$ is 1-Lipschitz. As an alternative strategy, Gulrajani et al. (2017) replace weight clipping with a gradient penalty (WGAN-GP) that also enforces the constraint. They demonstrate that their WGAN-GP strategy can successfully train a variety of model configurations where other GAN losses fail.

---

[1] Sound examples: `chrisdonahue.com/wavegan_examples`
Drum demo: `chrisdonahue.com/wavegan`
Generation with pre-trained models: `bit.ly/2G8NWpi`
Training code: `github.com/chrisdonahue/wavegan`

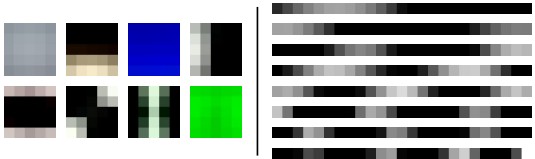

Figure 1: First eight principal components for 5x5 patches from natural images (**left**) versus those of length-25 audio slices from speech (**right**). Periodic patterns are unusual in natural images but a fundamental structure in audio.

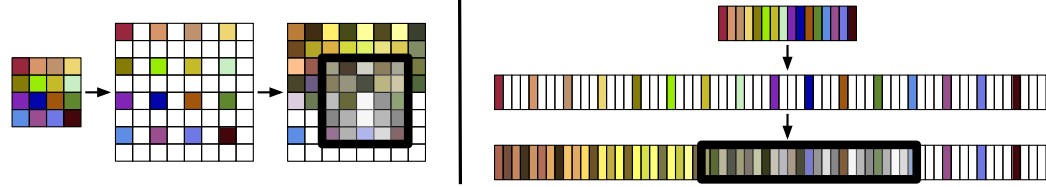

Figure 2: Depiction of the transposed convolution operation for the first layers of the DCGAN (Radford et al., 2016) (**left**) and WaveGAN (**right**) generators. DCGAN uses small (5x5), two-dimensional filters while WaveGAN uses longer (length-25), one-dimensional filters and a larger upsampling factor. Both strategies have the same number of parameters and numerical operations.

## 3 WAVEGAN

We motivate our design choices for WaveGAN by first highlighting the different types of structure found in audio versus images.

### 3.1 INTRINSIC DIFFERENCES BETWEEN AUDIO AND IMAGES

One way to illustrate the differences between audio and images is by examining the axes along which these types of data vary most substantially, i.e. by principal component analysis. In Figure 1, we show the first eight principal components for patches from natural images and slices from speech. While the principal components of images generally capture intensity, gradient, and edge characteristics, those from audio form a periodic basis that decompose the audio into constituent frequency bands. In general, natural audio signals are more likely to exhibit periodicity than natural images.

As a consequence, correlations across large windows are commonplace in audio. For example, in a waveform sampled at $16\,\text{kHz}$, a $440\,\text{Hz}$ sinusoid (the musical note A4) takes over 36 samples to complete a single cycle. This suggests that filters with larger receptive fields are needed to process raw audio. This same intuition motivated van den Oord et al. (2016) in their design of WaveNet, which uses dilated convolutions to exponentially increase the model's effective receptive field with linear increase in layer depth.

### 3.2 WAVEGAN ARCHITECTURE

We base our WaveGAN architecture off of DCGAN (Radford et al., 2016) which popularized usage of GANs for image synthesis. The DCGAN generator uses the *transposed convolution* operation (Figure 2) to iteratively upsample low-resolution feature maps into a high-resolution image. Motivated by our above discussion, we modify this transposed convolution operation to widen its receptive field. Specifically, we use longer one-dimensional filters of length 25 instead of two-dimensional filters of size 5x5, and we upsample by a factor of 4 instead of 2 at each layer (Figure 2). We modify the discriminator in a similar way, using length-25 filters in one dimension and increasing stride from 2 to 4. These changes result in WaveGAN having the same number of parameters, numerical operations, and output dimensionality as DCGAN.

Because DCGAN outputs 64x64 pixel images — equivalent to just 4096 audio samples — we add one additional layer to the model resulting in 16384 samples, slightly more than one second of

audio at $16\,\mathrm{kHz}$. This length is already sufficient for certain sound domains (e.g. sound effects, voice commands), and future work adapting megapixel image generation techniques (Karras et al., 2018) could expand the output length to more than a minute. We requantize the real data from its 16-bit integer representation (linear pulse code modulation) to 32-bit floating point, and our generator similarly outputs floating point waveforms. A complete description of our model is in Appendix D.

In summary, we outline our modifications to the DCGAN (Radford et al., 2016) method which result in WaveGAN. This straightforward recipe already produces reasonable audio, and further contributions outlined below and in Appendix A serve to refine results.

1. Flatten 2D convolutions into 1D (e.g. 5x5 2D convolution becomes length-25 1D).
2. Increase the stride factor for all convolutions (e.g. stride 2x2 becomes stride 4).
3. Remove batch normalization from the generator and discriminator.
4. Train using the WGAN-GP (Gulrajani et al., 2017) strategy.

### 3.3 PHASE SHUFFLE

Generative image models that upsample by transposed convolution (such as DCGAN) are known to produce characteristic "checkerboard" artifacts in images (Odena et al., 2016). Periodic patterns are less common in images (Section 3.1), and thus the discriminator can learn to reject images that contain them. For audio, analogous artifacts are perceived as pitched noise which may overlap with frequencies commonplace in the real data, making the discriminator's objective more challenging. However, the artifact frequencies will always occur at a particular phase, allowing the discriminator to learn a trivial policy to reject generated examples. This may inhibit the overall optimization problem.

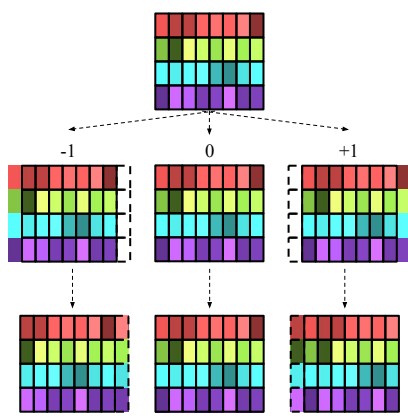

To prevent the discriminator from learning such a solution, we propose the *phase shuffle* operation with hyperparameter $n$. Phase shuffle randomly perturbs the phase of each layer's activations by $-n$ to $n$ samples before input to the next layer (Figure 3). We apply phase shuffle only to the discriminator, as the latent vector already provides the generator a mechanism to manipulate the phase of a resultant waveform. Intuitively speaking, phase shuffle makes the discriminator's job more challenging by requiring invariance to the phase of the input waveform.

Figure 3: At each layer of the Wave-GAN discriminator, the phase shuffle operation perturbs the phase of each feature map by Uniform $\sim [-n, n]$ samples, filling in the missing samples (dashed outlines) by reflection. Here we depict all possible outcomes for a layer with four feature maps ($n = 1$).

## 4 SPECGAN: GENERATING SEMI-INVERTIBLE SPECTROGRAMS

While a minority of recent research in discriminative audio classification tasks has used raw audio input (Sainath et al., 2015; Lee et al., 2017), most of these approaches operate on spectrogram representations of audio. A generative model may also benefit from operating in such a time-frequency space. However, commonly-used representations in the discriminative setting are uninvertible.

With SpecGAN, our frequency-domain audio generation model, we design a spectrogram representation that is both well-suited to GANs designed for image generation and *can* be approximately inverted. Additionally, to facilitate direct comparison, our representation is designed to use the same dimensionality per unit of time as WaveGAN (16384 samples yield a 128x128 spectrogram).

To process audio into suitable spectrograms, we first perform the short-time Fourier transform with $16\,\mathrm{ms}$ windows and $8\,\mathrm{ms}$ stride, resulting in 128 frequency bins[2] linearly spaced from 0 to $8\,\mathrm{kHz}$. We

---

[2]The FFT for this window size actually produces 129 frequency bins. We discard the top (Nyquist) bin from each example for training. During resynthesis, we replace it with the dataset's mean for that bin.

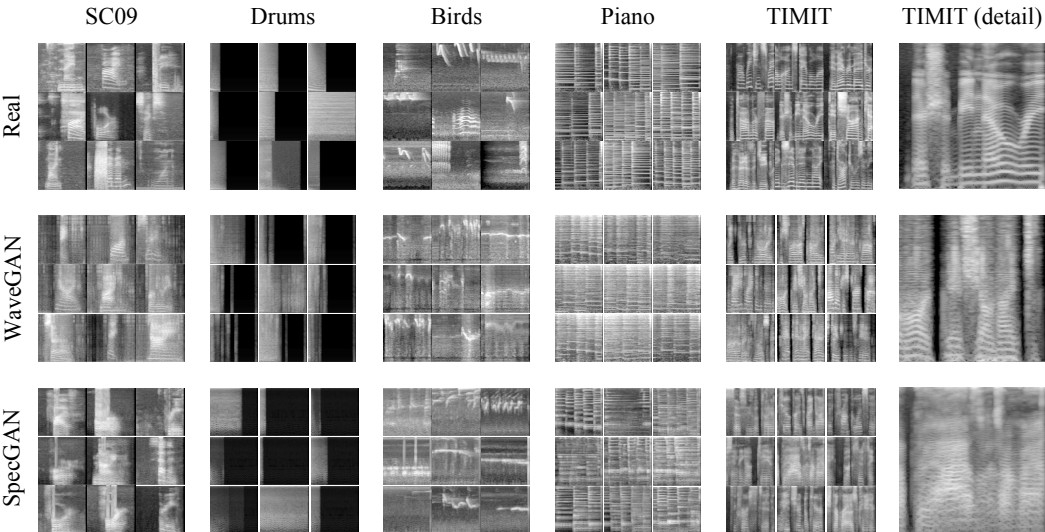

Figure 4: **Top**: Random samples from each of the five datasets used in this study, illustrating the wide variety of spectral characteristics. **Middle**: Random samples generated by WaveGAN for each domain. WaveGAN operates in the time domain but results are displayed here in the frequency domain for visual comparison. **Bottom**: Random samples generated by SpecGAN for each domain.

take the magnitude of the resultant spectra and scale amplitude values logarithmically to better-align with human perception. We then normalize each frequency bin to have zero mean and unit variance. This type of preprocessing is commonplace in audio classification, but produce spectrograms with unbounded values—a departure from image representations. We therefore clip the spectra to 3 standard deviations and rescale to $[-1, 1]$. Through an informal listening test, we determined that this clipping strategy did not produce an audible difference during inversion.

Once our dataset has been processed into this format, we operate the DCGAN (Radford et al., 2016) algorithm on the resultant spectra. To render the resultant generated spectrograms as waveforms, we first invert the steps of spectrogram preprocessing described above, resulting in linear-amplitude magnitude spectra. We then employ the iterative Griffin-Lim algorithm (Griffin & Lim, 1984) with 16 iterations to estimate phase and produce 16384 audio samples.

## 5 EXPERIMENTAL PROTOCOL

To facilitate human evaluation, our experimentation focuses on the *Speech Commands Dataset* (Warden, 2018). This dataset consists of many speakers recording individual words in uncontrolled recording conditions. We explore a subset consisting of the spoken digits "zero" through "nine" and refer to this subset as the *Speech Commands Zero Through Nine* (SC09) dataset. While this dataset is intentionally reminiscent of the popular MNIST dataset of written digits, we note that examples from SC09 are much higher dimensional ($\mathbb{R}^{16000}$) than examples from MNIST ($\mathbb{R}^{28 \times 28 = 784}$).

These ten words encompass many phonemes and two consist of multiple syllables. Each recording is one second in length, and we do not attempt to align the words in time. There are 1850 utterances of each word in the training set, resulting in 5.3 hours of speech. The heterogeneity of alignments, speakers, and recording conditions make this a challenging dataset for generative modeling.

Our baseline configuration for WaveGAN excludes phase shuffle. We compare this to the performance of WaveGAN with phase shuffle ($n \in \{2, 4\}$) and a variant of WaveGAN which uses nearest-neighbor upsampling rather than transposed convolution (Odena et al., 2016). Hoping to reduce noisy artifacts, we also experiment with adding a wide (length-512) post-processing filter to the output of the generator and learning its parameters with the rest of the generator variables (details in Appendix A.1). We use the WGAN-GP (Gulrajani et al., 2017) algorithm for all experiments, find-

ing it to produce reasonable results where others (Radford et al., 2016; Mao et al., 2017; Arjovsky et al., 2017) failed. We compare the performance of these configurations to that of SpecGAN.

We also perform experiments on four other datasets with different characteristics (Figure 4):

1. *Drum sound effects* (0.7 hours): Drum samples for kicks, snares, toms, and cymbals
2. *Bird vocalizations* (12.2 hours): In-the-wild recordings of many species (Boesman, 2018)
3. *Piano* (0.3 hours): Professional performer playing a variety of Bach compositions
4. *Large vocab speech (TIMIT)* (2.4 hours): Multiple speakers, clean (Garofolo et al., 1993)

We train our networks using batches of size 64 on a single NVIDIA P100 GPU. During our quantitative evaluation of SC09 (discussed below), our WaveGAN networks converge by their early stopping criteria (inception score) within four days (200k iterations, around 3500 epochs), and produce speech-like audio within the first hour of training. Our SpecGAN networks converge more quickly, within two days (around 1750 epochs). On the other four datasets, we train WaveGAN for 200k iterations representing nearly 1500 epochs for the largest dataset. Unlike with autoregressive methods (van den Oord et al., 2016; Mehri et al., 2017), generation with WaveGAN is fully parallel and can produce an hour of audio in less than two seconds. We list all hyperparameters in Appendix E.

## 6 EVALUATION METHODOLOGY

Evaluation of generative models is a fraught topic. Theis et al. (2016) demonstrate that quantitative measures of sample quality are poorly correlated with each other and human judgement. Accordingly, we use several quantitative evaluation metrics for hyperparameter validation and discussion, and also evaluate our most promising models with human judges.

### 6.1 INCEPTION SCORE

Salimans et al. (2016) propose the *inception score*, which uses a pre-trained Inception classifier (Szegedy et al., 2016) to measure both the diversity and semantic discriminability of generated images, finding that the measure correlates well with human judgement.

Given model scores $P(\boldsymbol{y} \mid \boldsymbol{x})$ with marginal $P(\boldsymbol{y})$, the inception score is defined as $\exp(\mathbb{E}_{\boldsymbol{x}} D_{\mathrm{KL}}(P(\boldsymbol{y} \mid \boldsymbol{x}) \| P(\boldsymbol{y})))$, and is estimated over a large number of samples (e.g. 50k). For $n$ classes, this measure ranges from 1 to $n$, and is maximized when the model is completely confident about each prediction *and* predicts each label equally often. We will use this measure as our primary quantitative evaluation method and early stopping criteria.

To measure inception score, we train an audio classifier on SC09. Our classifier first computes a short-time Fourier transform of the input audio with $64\,\mathrm{ms}$ windows and $8\,\mathrm{ms}$ stride. This representation is projected to 128 frequency bins equally spaced on the Mel scale (Stevens et al., 1937) from $40\,\mathrm{Hz}$ to $7800\,\mathrm{Hz}$. Amplitudes are scaled logarithmically and normalized so that each bin has zero mean and unit variance. We process this perceptually-informed representation with four layers of convolution and pooling, projecting the result to a *softmax* layer with 10 classes. We perform early stopping on the minimum negative log-likelihood of the validation set; the resultant model achieves $93\%$ accuracy on the test set. Because this classifier observes spectrograms, our spectrogram-generating models may have a representational advantage over our waveform-generating models.

### 6.2 NEAREST NEIGHBOR COMPARISONS

Inception score has two trivial failure cases in which a poor generative model can achieve a high score. Firstly, a generative model that outputs a single example of each class with uniform probability will be assigned a high score. Secondly, a generative model that overfits the training data will achieve a high score simply by outputting examples on which the classifier was trained.

We use two indicators metrics to determine if a high inception score has been caused by either of these two undesirable cases. Our first indicator, $|D|_{\mathrm{self}}$, measures the average Euclidean distance of a set of 1k examples to their nearest neighbor within the set (other than itself). A higher $|D|_{\mathrm{self}}$ indicates higher diversity amongst samples. Because measuring Euclidean distance in time-domain

Table 1: Quantitative and qualitative (human study) results for SC09 experiments comparing real and generated data. A higher inception score suggests that semantic modes of the real data distribution have been captured. $|D|_{\text{self}}$ indicates the intra-dataset diversity relative to that of the real test data. $|D|_{\text{train}}$ indicates the distance between the dataset and the training set relative to that of the test data; a low value indicates a generative model that is overfit to the training data. Acc. is the overall accuracy of humans on the task of labeling class-balanced digits (random chance is 0.1). Sound *quality*, *ease* of intelligibility and speaker *diversity* are mean opinion scores (1-5); higher is better.

| Experiment | Quantitative | | | Qualitative (human judges) | | | |
| --- | --- | --- | --- | --- | --- | --- | --- |
| | Inception score | $|D|_{\text{self}}$ | $|D|_{\text{train}}$ | Acc. | Quality | Ease | Diversity |
| Real (train) | $9.18 \pm 0.04$ | 1.1 | 0.0 | | | | |
| Real (test) | $8.01 \pm 0.24$ | 1.0 | 1.0 | 0.95 | $3.9 \pm 0.8$ | $3.9 \pm 1.1$ | $3.5 \pm 1.0$ |
| Parametric | $5.02 \pm 0.06$ | 0.7 | 1.1 | | | | |
| WaveGAN | $4.12 \pm 0.03$ | 1.4 | 2.0 | | | | |
| + Phase shuffle $n = 2$ | $4.67 \pm 0.01$ | 0.8 | 2.3 | 0.58 | $2.3 \pm 0.9$ | $2.8 \pm 0.9$ | $3.2 \pm 0.9$ |
| + Phase shuffle $n = 4$ | $4.54 \pm 0.03$ | 1.0 | 2.3 | | | | |
| + Nearest neighbor | $3.77 \pm 0.02$ | 1.8 | 2.6 | | | | |
| + Post-processing | $3.92 \pm 0.03$ | 1.4 | 2.9 | | | | |
| + Dropout | $3.93 \pm 0.03$ | 1.0 | 2.6 | | | | |
| SpecGAN | $6.03 \pm 0.04$ | 1.1 | 1.4 | 0.66 | $1.9 \pm 0.8$ | $2.8 \pm 0.9$ | $2.6 \pm 1.0$ |
| + Phase shuffle $n = 1$ | $3.71 \pm 0.03$ | 0.8 | 1.6 | | | | |

audio poorly represents human perception, we evaluate distances in the same frequency-domain representation as our classifier from Section 6.1.

Our second indicator, $|D|_{\text{train}}$, measures the average Euclidean distance of 1k examples to their nearest neighbor in the real training data. If the generative model simply produces examples from the training set, this measure will be 0. We report $|D|_{\text{train}}$ and $|D|_{\text{self}}$ relative to those of the test set.

### 6.3 Qualitative human judgements

While inception score is a useful metric for hyperparameter validation, our ultimate goal is to produce examples that are intelligible to humans. To this end, we measure the ability of human annotators on *Amazon Mechanical Turk* to label the generated audio. Using our best WaveGAN and SpecGAN models as measured by inception score, we generate random examples until we have 300 for each digit (as labeled by our classifier from Section 6.1)—3000 total. In batches of ten random examples, we ask annotators to label which digit they perceive in each example, and compute their accuracy with respect to the classifier's labels (random accuracy would be $10\%$). After each batch, annotators assign subjective values of 1–5 for criteria of sound quality, ease of intelligibility, and speaker diversity. We report accuracy ($n = 3000$) and mean opinion scores ($n = 300$) in Table 1.

## 7 Results and discussion

Results for our evaluation appear in Table 1. We also evaluate our metrics on the real training data, the real test data, and a version of SC09 generated by a parametric speech synthesizer (Buchner, 2017). We also compare to SampleRNN (Mehri et al., 2017) and two public implementations of WaveNet (van den Oord et al., 2016), but neither method produced competitive results (details in Appendix B), and we excluded them from further evaluation. These autoregressive models have not previously been examined on small-vocabulary speech data, and their success at generating full words has only been demonstrated when conditioning on rich linguistic features. Sound examples for all experiments can be found at chrisdonahue.com/wavegan_examples.

While the maximum inception score for SC09 is 10, any score higher than the test set score of 8 should be seen as evidence that a generative model has overfit. Our best WaveGAN model uses phase shuffle with $n = 2$ and achieves an inception score of 4.7. To compare the effect of phase shuffle to other common regularizers, we also tried using $50\%$ dropout in the discriminator's activations, which

resulted in a lower score. Phase shuffle decreased the inception score of SpecGAN, possibly because the operation has an exaggerated effect when applied to the compact temporal axis of spectrograms.

Most experiments produced $|D|_{\text{self}}$ (diversity) values higher than that of the test data, and all experiments produced $|D|_{\text{train}}$ (distance from training data) values higher than that of the test data. While these measures indicate that our generative models produce examples with statistics that deviate from those of the real data, neither metric indicates that the models achieve high inception scores by the trivial solutions outlined in Section 6.2.

Compared to examples from WaveGAN, examples from SpecGAN achieve higher inception score ($6.0$ vs. $4.7$) and are labeled more accurately by humans ($66\%$ vs. $58\%$). However, on subjective criteria of sound quality and speaker diversity, humans indicate a preference for examples from WaveGAN. It appears that SpecGAN might better capture the variance in the underlying data compared to WaveGAN, but its success is compromised by sound quality issues when its spectrograms are inverted to audio. It is possible that the poor qualitative ratings for examples from SpecGAN are primarily caused by the lossy Griffin-Lim inversion (Griffin & Lim, 1984) and not the generative procedure itself. We see promise in both waveform and spectrogram audio generation with GANs; our study does not suggest a decisive winner. For a more thorough investigation of spectrogram generation methods, we point to follow-up work (Engel et al., 2019).

Finally, we train WaveGAN and SpecGAN models on the four other domains listed in Section 5. Somewhat surprisingly, we find that the frequency-domain spectra produced by WaveGAN (a time-domain method) are visually more consistent with the training data (e.g. in terms of sharpness) than those produced by SpecGAN (Figure 4). For drum sound effects, WaveGAN captures semantic modes such as kick and snare drums. On bird vocalizations, WaveGAN generates a variety of distinct bird sounds. On piano, WaveGAN produces musically-consonant motifs that, as with the training data, represent a variety of key signatures and rhythmic patterns. For TIMIT, a large-vocabulary speech dataset with many speakers, WaveGAN produces speech-like babbling similar to results from unconditional autoregressive models (van den Oord et al., 2016).

# 8 RELATED WORK

Much of the work within generative modeling of audio is within the context of text-to-speech. Text-to-speech systems are primarily either *concatenative* or *parametric*. In concatenative systems, audio is generated by sequencing small, prerecorded portions of speech from a phonetically-indexed dictionary (Moulines & Charpentier, 1990; Hunt & Black, 1996). Parametric systems map text to salient parameters of speech, which are then synthesized by a vocoder (Dudley, 1939); see (Zen et al., 2009) for a comprehensive review. Some of these systems use learning-based approaches such as a hidden Markov models (Yoshimura, 2002; Tokuda et al., 2013), and separately-trained neural networks pipelines (Ling et al., 2015) to estimate speech parameters.

Recently, several researchers have investigated parametric speech synthesis with end-to-end neural network approaches that learn to produce vocoder features directly from text or phonetic embeddings (Arik et al., 2017; Ping et al., 2018; Sotelo et al., 2017; Wang et al., 2017; Shen et al., 2018). These vocoder features are synthesized to raw audio using off-the-shelf methods such as WORLD (Morise et al., 2016) and Griffin-Lim (Griffin & Lim, 1984), or trained neural vocoders (Sotelo et al., 2017; Shen et al., 2018; Ping et al., 2018). All of these methods are supervised: they are trained to map linguistic features to audio outputs.

Several approaches have explored unsupervised generation of raw audio. van den Oord et al. (2016) propose WaveNet, a convolutional model which learns to predict raw audio samples by autoregressive modeling. WaveNets conditioned on rich linguistic features have widely been deployed in text-to-speech systems, though they have not been demonstrated capable of generating cohesive words in the unconditional setting. Engel et al. (2017) pose WaveNet as an autoencoder to generate musical instrument sounds. Chung et al. (2014); Mehri et al. (2017) both train recurrent autoregressive models which learn to predict raw audio samples. While autoregressive methods generally produce higher audio fidelity than WaveGAN, synthesis with WaveGAN is orders of magnitude faster.

The application of GANs (Goodfellow et al., 2014) to audio has so far been limited to supervised learning problems in combination with traditional loss functions. Pascual et al. (2017) apply GANs to raw audio speech enhancement. Their encoder-decoder approach combines the GAN objective

with an $L_2$ loss. Fan et al. (2017); Michelsanti & Tan (2017); Donahue et al. (2018) all use GANs in combination with unstructured losses to map spectrograms in one domain to spectrograms in another. Chen et al. (2017) use GANs to map musical performance images into spectrograms.

# 9   CONCLUSION

We present WaveGAN, the first application of GANs to unsupervised audio generation. WaveGAN is fully parallelizable and can generate hours of audio in only a few seconds. In its current form, WaveGAN can be used for creative sound design in multimedia production. In our future work we plan to extend WaveGAN to operate on variable-length audio and also explore a variety of label conditioning strategies. By providing a template for modifying image generation models to operate on audio, we hope that this work catalyzes future investigation of GANs for audio synthesis.

### ACKNOWLEDGMENTS

The authors would like to thank Peter Boesman and Colin Raffel for providing training data for this work. This work was supported by the Unity Global Graduate Fellowship program and the UC San Diego Department of Computer Science. GPUs used for this work were provided by the HPC @ UC program and donations from NVIDIA.

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

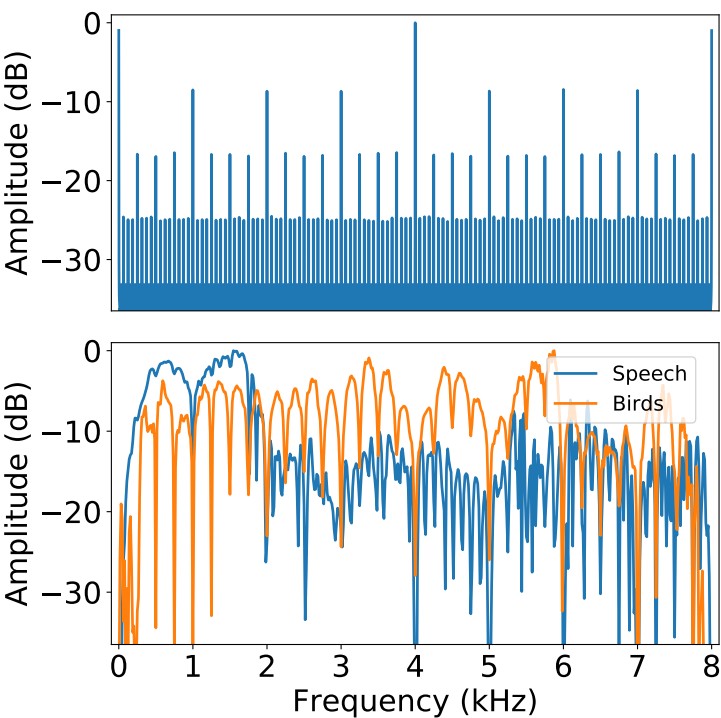

Figure 5: (**Top**): Average impulse response for 1000 random initializations of the WaveGAN generator. (**Bottom**): Response of learned post-processing filters for speech and bird vocalizations. Post-processing filters reject frequencies corresponding to noise byproducts created by the generative procedure (top). The filter for speech boosts signal in prominent speech bands, while the filter for bird vocalizations (which are more uniformly-distributed in frequency) simply reduces noise presence.

## A   UNDERSTANDING AND MITIGATING ARTIFACTS IN GENERATED AUDIO

Generative models that upsample by transposed convolution are known to produce characteristic "checkerboard" artifacts in images (Odena et al., 2016), artifacts with particular spatial periodicities. The discriminator of image-generating GANs can learn to reject images with these artifacts because they are uncommon in real data (as discussed in Section 3.1). However, in the audio domain, the discriminator might not have such luxury as these artifacts correspond to frequencies which might rightfully appear in the real data.

While checkerboard artifacts are an annoyance in image generation, they can be devastating to audio generation results. While our eye may perceive these types of periodic distortions as an intrusive texture, our ear perceives them as an abrasive tone. To characterize these artifacts in WaveGAN, we measure its impulse response by randomly initializing it 1000 times and passing unit impulses to its first convolutional layer. In Figure 5, we plot the average of these responses in the frequency domain. The response has sharp peaks at linear multiples of the sample rates of each convolutional layer ($250\,\mathrm{Hz}$, $1\,\mathrm{kHz}$, $4\,\mathrm{kHz}$, etc.). This is in agreement with our informal observation of results from WaveGAN, which often have a pitched noise close to the musical note B ($247 \times 2^n$ Hz).

Below, we will discuss strategies we designed to mitigate these artifacts in WaveGAN.

### A.1   LEARNED POST-PROCESSING FILTERS

We experiment with adding a post-processing filter to the generator, giving WaveGAN a simple mechanism to filter out undesirable frequencies created by the generative process. This filter has a long window (512 samples) allowing it to represent intricate transfer functions, and the weights

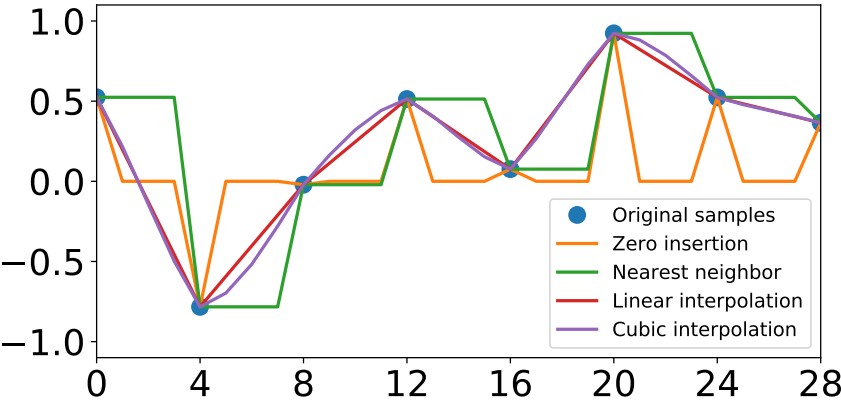

Figure 6: Depiction of the upsampling strategy used by transposed convolution (zero insertion) and other strategies which mitigate aliasing: nearest neighbor, linear and cubic interpolation.

of the filter are learned as part of the generator's parameters. In Figure 5, we compare the post-processing filters that WaveGAN learns for human speech and bird vocalizations. The filters boost signal in regions of the frequency spectrum that are most prominent in the real data domain, and introduce notches at bands that are artifacts of the generative procedure as discussed in the previous section.

### A.2 UPSAMPLING PROCEDURE

Transposed convolution upsamples signals by inserting zeros in between samples and applying a learned filterbank. This operation introduces aliased frequencies, copies of pre-existing frequencies shifted by multiples of the new Nyquist rate, into the upsampled signal. While aliased frequencies are usually seen as undesirable artifacts of a bad upsampling procedure, in the generative setting their existence may be crucial for producing fine-grained details in the output.

We experiment with three other upsampling strategies in WaveGAN: nearest-neighbor, linear and cubic interpolation, all of which attenuate aliased frequencies. In Figure 6, we compare these strategies visually. While nearest neighbor upsampling resulted in similar audio output to transposed convolution, linear and cubic interpolation strategies resulted in qualitatively poor audio output (sound examples: `chrisdonahue.com/wavegan_examples`). We hypothesize that the aliased frequencies produced by upsampling convolutions may be more critical to audio generation than image generation.

### B EXPERIMENTS WITH AUTOREGRESSIVE WAVEFORM MODELS

We developed our WaveGAN and SpecGAN models primarily to address the task of steerable sound effect generation. This is an inherently different task than text to speech (TTS), however autoregressive waveform models (e.g. WaveNet (van den Oord et al., 2016) and SampleRNN (Mehri et al., 2017)) that were developed for TTS can also be used to model and generate waveforms unconditionally. Hence, a comparison to these models for our task is reasonable. One upside of autoregressive models for our task is that they have the potential to produce high-quality audio. Potential downsides are 1) these models take several orders of magnitude longer to generate waveforms, and 2) they do not learn a compact latent space of waveforms, causing useful sound generation tasks like continuous exploration and interpolation to be impossible.

We attempt to train **two** public implementations of WaveNet (ImplA[3] and ImplB[4]) and SampleRNN[5] on our SC09 digit generation task. We use default parameters for these libraries; the only modifications we make are to reduce the training example size to one second. To our ears, all three libraries

---

[3]WaveNet ImplA (unofficial): `github.com/ibab/tensorflow-wavenet`
[4]WaveNet ImplB (unofficial): `github.com/r9y9/wavenet_vocoder`
[5]SampleRNN (official): `github.com/soroushmehr/sampleRNN_ICLR2017`

failed to produce cohesive words (you can judge for yourself from our sound examples at the bottom `chrisdonahue.com/wavegan_examples`). This poor subjective performance is echoed by weak inception scores (weaker than any in Table 1): $1.07 \pm 0.05$, $1.29 \pm 0.03$, $2.28 \pm 0.19$ for WaveNet ImplA, WaveNet ImplB, and SampleRNN respectively. Note that these inception scores were calculated on far fewer examples ($< 1k$) than all of the scores listed in Table 1 (which were computed on $50k$ examples). This is because it took over $24$ hours to produce even a thousand one-second examples with these methods (whereas our methods produce $50k$ examples in a few seconds).

Autoregressive methods have not been demonstrated capable of learning to synthesize coherent words without conditioning on rich linguistic features. We are not claiming that these methods *cannot* learn to synthesize full words, merely that three open-source implementations were unable to do so with default parameters. We want to be clear that our intent is not to disparage autoregressive waveform methods as these methods were developed for a different task, and hence we excluded these poor scores from our results table to avoid sending the wrong message. Instead, we hope to highlight that these implementations produced results that were noncompetitive for our problem domain, and less useful (due to slowness and lack of a latent space) for creative generation of sound effects.

## C  FELINE "TURING TEST"

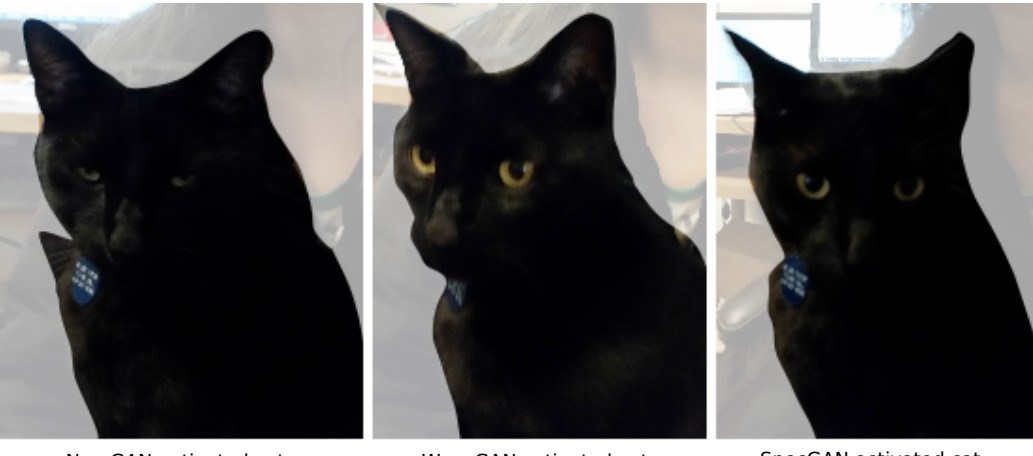

| Non GAN-activated cat | WaveGAN activated cat | SpecGAN activated cat |

Figure 7: Compared to resting state, this cat's level of alertness increased when presented bird vocalizations synthesized by WaveGAN and SpecGAN.

As our results improved throughout the course of this research, our cats became quite intrigued by the synthetic bird vocalizations produced by WaveGAN (Figure 7). While this was of course not a formal experiment, we did find this to be encouraging evidence that our method might be capable of producing audio that could additionally convince non-human animals.

Table 2: WaveGAN generator architecture

| Operation | Kernel Size | Output Shape |
|---|---|---|
| Input $z \sim \text{Uniform}(-1, 1)$ | | $(n, 100)$ |
| Dense 1 | $(100, 256d)$ | $(n, 256d)$ |
| Reshape | | $(n, 16, 16d)$ |
| ReLU | | $(n, 16, 16d)$ |
| Trans Conv1D (Stride=4) | $(25, 16d, 8d)$ | $(n, 64, 8d)$ |
| ReLU | | $(n, 64, 8d)$ |
| Trans Conv1D (Stride=4) | $(25, 8d, 4d)$ | $(n, 256, 4d)$ |
| ReLU | | $(n, 256, 4d)$ |
| Trans Conv1D (Stride=4) | $(25, 4d, 2d)$ | $(n, 1024, 2d)$ |
| ReLU | | $(n, 1024, 2d)$ |
| Trans Conv1D (Stride=4) | $(25, 2d, d)$ | $(n, 4096, d)$ |
| ReLU | | $(n, 4096, d)$ |
| Trans Conv1D (Stride=4) | $(25, d, c)$ | $(n, 16384, c)$ |
| Tanh | | $(n, 16384, c)$ |

Table 3: WaveGAN discriminator architecture

| Operation | Kernel Size | Output Shape |
|---|---|---|
| Input $x$ or $G(z)$ | | $(n, 16384, c)$ |
| Conv1D (Stride=4) | $(25, c, d)$ | $(n, 4096, d)$ |
| LReLU ($\alpha = 0.2$) | | $(n, 4096, d)$ |
| Phase Shuffle ($n = 2$) | | $(n, 4096, d)$ |
| Conv1D (Stride=4) | $(25, d, 2d)$ | $(n, 1024, 2d)$ |
| LReLU ($\alpha = 0.2$) | | $(n, 1024, 2d)$ |
| Phase Shuffle ($n = 2$) | | $(n, 1024, 2d)$ |
| Conv1D (Stride=4) | $(25, 2d, 4d)$ | $(n, 256, 4d)$ |
| LReLU ($\alpha = 0.2$) | | $(n, 256, 4d)$ |
| Phase Shuffle ($n = 2$) | | $(n, 256, 4d)$ |
| Conv1D (Stride=4) | $(25, 4d, 8d)$ | $(n, 64, 8d)$ |
| LReLU ($\alpha = 0.2$) | | $(n, 64, 8d)$ |
| Phase Shuffle ($n = 2$) | | $(n, 64, 8d)$ |
| Conv1D (Stride=4) | $(25, 8d, 16d)$ | $(n, 16, 16d)$ |
| LReLU ($\alpha = 0.2$) | | $(n, 16, 16d)$ |
| Reshape | | $(n, 256d)$ |
| Dense | $(256d, 1)$ | $(n, 1)$ |

## D   ARCHITECTURE DESCRIPTION

In Tables 2 and 3, we list the full architectures for our WaveGAN generator and discriminator respectively. In Tables 4 and 5, we list the same for SpecGAN. In these tables, $n$ is the batch size, $d$ modifies model size, and $c$ is the number of channels in the examples. In all of our experiments in this paper, $c = 1$. All dense and convolutional layers include biases. No batch normalization is used in WaveGAN or SpecGAN.

## E   TRAINING HYPERPARAMETERS

In Table 6, we list the values of these and all other hyperparameters for our experiments, which constitute our out-of-the-box recommendations for WaveGAN and SpecGAN.

Table 4: SpecGAN generator architecture

| Operation | Kernel Size | Output Shape |
|---|---|---|
| Input $z \sim \text{Uniform}(-1, 1)$ | | $(n, 100)$ |
| Dense 1 | $(100, 256d)$ | $(n, 256d)$ |
| Reshape | | $(n, 4, 4, 16d)$ |
| ReLU | | $(n, 4, 4, 16d)$ |
| Trans Conv2D (Stride=2) | $(5, 5, 16d, 8d)$ | $(n, 8, 8, 8d)$ |
| ReLU | | $(n, 8, 8, 8d)$ |
| Trans Conv2D (Stride=2) | $(5, 5, 8d, 4d)$ | $(n, 16, 16, 4d)$ |
| ReLU | | $(n, 16, 16, 4d)$ |
| Trans Conv2D (Stride=2) | $(5, 5, 4d, 2d)$ | $(n, 32, 32, 2d)$ |
| ReLU | | $(n, 32, 32, 2d)$ |
| Trans Conv2D (Stride=2) | $(5, 5, 2d, d)$ | $(n, 64, 64, d)$ |
| ReLU | | $(n, 64, 64, d)$ |
| Trans Conv2D (Stride=2) | $(5, 5, d, c)$ | $(n, 128, 128, c)$ |
| Tanh | | $(n, 128, 128, c)$ |

Table 5: SpecGAN discriminator architecture

| Operation | Kernel Size | Output Shape |
|---|---|---|
| Input $x$ or $G(z)$ | | $(n, 128, 128, c)$ |
| Conv2D (Stride=2) | $(5, 5, c, d)$ | $(n, 64, 64, d)$ |
| LReLU ($\alpha = 0.2$) | | $(n, 64, 64, d)$ |
| Conv2D (Stride=2) | $(5, 5, d, 2d)$ | $(n, 32, 32, 2d)$ |
| LReLU ($\alpha = 0.2$) | | $(n, 32, 32, 2d)$ |
| Conv2D (Stride=2) | $(5, 5, 2d, 4d)$ | $(n, 16, 16, 4d)$ |
| LReLU ($\alpha = 0.2$) | | $(n, 16, 16, 4d)$ |
| Conv2D (Stride=2) | $(5, 5, 4d, 8d)$ | $(n, 8, 8, 8d)$ |
| LReLU ($\alpha = 0.2$) | | $(n, 8, 8, 8d)$ |
| Conv2D (Stride=2) | $(5, 5, 8d, 16d)$ | $(n, 4, 4, 16d)$ |
| LReLU ($\alpha = 0.2$) | | $(n, 4, 4, 16d)$ |
| Reshape | | $(n, 256d)$ |
| Dense | $(256d, 1)$ | $(n, 1)$ |

Table 6: WaveGAN hyperparameters

| Name | Value |
|---|---|
| Input data type | 16-bit PCM (requantized to 32-bit float) |
| Model data type | 32-bit floating point |
| Num channels ($c$) | 1 |
| Batch size ($b$) | 64 |
| Model dimensionality ($d$) | 64 |
| Phase shuffle (WaveGAN) | 2 |
| Phase shuffle (SpecGAN) | 0 |
| Loss | WGAN-GP (Gulrajani et al., 2017) |
| WGAN-GP $\lambda$ | 10 |
| $D$ updates per $G$ update | 5 |
| Optimizer | Adam ($\alpha = 1e{-}4$, $\beta_1 = 0.5$, $\beta_2 = 0.9$) |

