# OpenReview forum: "Adversarial Audio Synthesis"
_ICLR.cc/2019/Conference_

### Official Review · AnonReviewer1 · 2018-11-04
**Review - Adversarial Audio Synthesis**

**Rating:** 6
**Confidence:** 4

**Review:**



*Pros:*
-	Easily accessible paper with good illustrations and a mostly fair presentation of the results (see suggestions below).
-	It is a first attempt to generate audio with GANs which results in an efficient scheme for generating short, fixed-length audio segments of reasonable (but not high) quality.
-	Human evaluations (using crowdsourcing) provides empirical evidence that the approach has merit.
-	The paper appears reproducible and comes with data and code.

*Cons*:
-	Potentially a missing comparison with existing generative methods (e.g. WaveNet). See comments/questions below **
-	The underlying idea is relatively straightforward in that the proposed methods is a non-trivial application of already known techniques from ML and audio signal processing.

*Significance*: The proposed GAN-based audio generator is an interesting step in the development of more efficient audio generation and it is of interest to a subcommunity of ICLR as it provides a number of concrete techniques for applying GANs to audio.

*Further comments/ questions:*
-	Abstract/introduction: I’d suggest being more explicit about the limitations of the method, i.e. you are currently able to generate short and fixed-length audio.
-	SpecGAN (p 4): I’d suggest including some justification of the chosen pre-processing of spectrograms (p. 4, last paragraph).
-	** Evaluation:  The paper dismisses existing generative methods early in the evaluation phase but the justification for doing so is not entirely clear to me: Firstly, if the inception score is used as an objective criterion it would seem reasonable to include the values in the paper. Secondly, as inception scores are based on spectrograms it could potentially favour methods using spectrograms directly (SpecGAN) or indirectly (WaveGAN, via early stopping) thus putting the purely sample based methods (e.g. WaveNet) at a disadvantage. It would seem fair to pre-screen the audio before dismissing competitors instead of solely relying on potentially biased inception scores (which was probably also done in this work, but not clearly stated…)? Finally, while not the aim of the paper, it would have been beneficial to discuss and understand the failures of existing methods in more detail to convince the reader that a fair attempt has been made to getting competitors to work before leaving them out entirely.
-	Results/analysis: It is unclear to me how many people annotated the individual samples? What is the standard deviation over the human responses (perhaps include in tab 1)? Consider including a reflection on (or perhaps even test statistically) the alignment between the qualitative diversity/quality scores and the subjective ratings to justify the use of the objective scores in the training/selection process.
-	Related work: I think it would provide a better narrative if the existing techniques are outlined earlier on in the paper.

---

> ### Author Response · Authors · 2018-11-08
> **Author Response to Reviewer #1**
>
> Thank you for your thoughtful comments and suggestions. We will respond to each of your points below.
>
> ** Explicit mention of methodological limitations **
>
> We have updated the abstract and introduction to clarify that our model produces fixed-length results. We added the following sentence to our abstract “WaveGAN is capable of synthesizing one second audio waveforms with temporal coherence, suitable for sound effect generation.” We also added a similar clarification to paragraph 5 of the introduction (specifying that the generated waveforms are one second in length).
>
> ** Justification for spectrogram pre-processing **
>
> We added justification for our spectrogram preprocessing to the last paragraph of page 4.
>
> ** Discussion of existing methods (e.g. WaveNet) **
>
> “““The paper dismisses existing generative methods early in the evaluation phase … it would have been beneficial to discuss and understand the failures of existing methods in more detail to convince the reader that a fair attempt has been made to getting competitors to work before leaving them out entirely”””
>
> We had originally included some of these details in our paper but they were cut for brevity. We agree that we cut too much, and have added details back into the paper in the form of a new Appendix section (Appendix C) with a pointer from the main paper. A summary follows:
>
> How autoregressive waveform models (e.g. WaveNet) factor into the story and evaluation of our paper is a tricky subject, and one that we tried to handle thoughtfully. First and foremost: *the two public implementations of WaveNet that we tried simply failed to produce reasonable results* (sound examples can be heard at the bottom of http://iclr-wavegan-results.s3-website-us-east-1.amazonaws.com ). We did informally pre-screen these results ourself (and you can as well) and concluded that they were clearly noncompetitive. We also calculated Inception scores for these experiments: they were 1.067 +- 0.045 and 1.293 +- 0.027 respectively.
>
> We reasoned that including these (poor) numbers in our results table would send the wrong message to readers. Namely, it would appear that we are claiming our methods works better than WaveNet. *This is NOT a claim that we are attempting to make*, as WaveNet was developed for a different problem (text-to-speech) than the one we are focusing on (learning the semantics of short audio clips). WaveNet additionally has no concept of a latent space, which would not allow for the same steerable exploration of sound effects that our model aspires to achieve (outlined in the introduction). Furthermore, we expect that the proprietary implementation of WaveNet would produce something more reasonable for our spoken digits task, but unfortunately we do not have access to it.
>
> ** User study clarification **
>
> “““ It is unclear to me how many people annotated the individual samples? ”””
>
> We have 300 examples of each digit, resulting in 3000 total labeling problems (name the digit 1-10). We give these to 300 annotators in random batches of 10 examples, and ask for qualitative assessments at the end of each batch. Accordingly, we have 300 responses to each qualitative metric (quality, easy, diversity). Standard deviations for MOS scores are around 1 for each category, resulting in small standard errors (~0.06) for n=300. We have added the standard deviations to our paper table and updated the text in Section 6.3 to clarify these details.
>
> “““ Consider including a reflection on (or perhaps even test statistically) the alignment between the qualitative diversity/quality scores and the subjective ratings to justify the use of the objective scores in the training/selection process ”””
>
> The evaluation of generative models is a fraught topic and the lack of correlation between quantitative and qualitative metrics is known (see “A note on the evaluation of generative models” Theis et al. ICLR 2016). In the scope of our work, we do not have enough data points (only three for the expensive Mechanical Turk evaluations) to reach substantive conclusions about the correlation between e.g. Inception score and mean opinion scores for quality.
>
> We hypothesize that the discrepancy between our quantitative metrics (Inception score, nearest neighbor comparisons) and subjective metrics (MOS scores) is due to the fact that the former are computed from spectrograms while the latter are from humans listening to waveforms. Unfortunately, evaluation of Inception score in the waveform domain was impractical as we were unable to train a waveform domain classifier that achieved reasonable accuracy on this classification task (note that we mention in our abstract that audio classifiers usually operate on spectrograms). However, we have updated our discussion to clarify: “This discrepancy can likely be attributed to the fact that inception scores are computed on spectrograms while subjective quality assessments are made by humans listening to waveforms.”

---

### Official Review · AnonReviewer2 · 2018-11-06
**This paper proposes WaveGAN for unsupervised synthesis of raw-wave-form audio**

**Rating:** 6
**Confidence:** 3

**Review:**

This paper proposes WaveGAN for unsupervised synthesis of raw-wave-form audio and SpecGAN that based on spectrogram. Experimental results look promising.

I still believe the goal should be developing a text-to-speech synthesizer, at least one aspect.

---

> ### Author Response · Authors · 2018-11-08
> **Author Response to Reviewer #2**
>
> Thank you for highlighting that our experimental results are promising. As you mentioned, we state in our paper that “though our evaluation focuses on a speech generation task, we note that it is not our goal to develop a text-to-speech synthesizer.” *We are primarily targeting generation of novel sound effects as our task.* We think this is an important task with immediate application to creative domains (e.g. music production, film scoring) and is orthogonal to the task of synthesizing realistic speech from transcripts. Our model is already capable of producing convincing results on this task for several different sound domains. Furthermore, whereas the goals for text to speech are to synthesize a given transcript, we are providing a method which enables user-driven content generation through exploration of a compact latent space of sound effects.
>
> Our purpose for focusing our evaluation on a speech generation task is to enable straightforward annotating for humans on Mechanical Turk. From the paper: “While our objective is sound effect generation (e.g. generating drum sounds), human evaluation for these tasks would require expert listeners. Therefore, we also consider a speech benchmark, facilitating straightforward assessment by human annotators.”

---

### Official Review · AnonReviewer3 · 2018-11-07
**Interesting application, limited algorithmic contribution**

**Rating:** 5
**Confidence:** 4

**Review:**

This paper applies GANs for unsupervised audio generation. Particularly, DCGAN-like models are applied for generating audio. This application is interesting, but the algorithmic contribution is limited.

Qualitative ratings are poor. The important problem of generating variable-length audio is untouched.

---

> ### Author Response · Authors · 2018-11-08
> **Author Response to Reviewer #3**
>
> Thank you for your feedback. We appreciate that you found our application to be interesting. We will address your criticisms in order.
>
> We noticed you changed your score from a 6 to a 5 without updating the text of your review. We would be happy to address your concerns if you can provide additional context as to the reasoning behind your rating change.
>
> “““ the algorithmic contribution is limited. ”””
>
> We would like to reiterate that our paper is the first to apply GANs to audio generation which is not as straightforward as simply adapting existing models. Specifically, we believe we have made concrete methodological contributions such as phase shuffle (Section 3.3) and the learned post processing filters (Appendix B). In particular, phase shuffle was observed to increase Inception scores substantially (4.1->4.7), and, to our ears, made the difference between spoken digits that were intelligible and those that were unintelligible.
>
> Spoken digits from WaveGAN **with** phase shuffle (more intelligible): http://iclr-wavegan-results.s3-website-us-east-1.amazonaws.com/quant_wavegan_ps2.wav
> Spoken digits from WaveGAN **without** phase shuffle (less intelligible): http://iclr-wavegan-results.s3-website-us-east-1.amazonaws.com/quant_wavegan.wav
>
> “““ Qualitative ratings are poor. ”””
>
> As our task seeks to evaluate how well GANs can capture the semantic modes (vocabulary words in this case) of the training data, the primary qualitative metric to pay attention to should be the labeling accuracy. We believe our results of around 60% accuracy for generated data show that our approach is promising (note that random chance would be 10%).
>
> On the subject of the qualitative ratings, our primary goal with this work is to provide a reasonable first pass at this problem, as well as define a task with clear and reproducible evaluation methodology to allow ourselves and others to iterate further. We believe our qualitative results are adequate, but note that improving these scores is a promising avenue for future work by integrating recent breakthroughs in image processing such as spectral normalization (Miyato et al. ICLR 2018) and progressive growth (Kerras et al. ICLR 2018).
>
> “““ The important problem of generating variable-length audio is untouched. ”””
>
> We were the first to tackle fixed-length audio generation with GANs, a task which is already useful for application in several creative domains that we mention in the paper (music production, film scoring). We hope to build on our results in future work to address the challenging problem of generating variable-length audio.

---

> > ### Comment · AnonReviewer3 · 2018-11-12
> > **More explanations on my scoring**
> >
> > Thanks for your clarifications. Here are my thoughts on modifying the score.
> >
> > “““ the algorithmic contribution is limited. ”””
> >
> > If as said in the response, the concrete methodological contributions are phase shuffle (Section 3.3) and the learned post processing filters (Appendix B). At first reading of the paper, these contributions are not clear. These statements are not presented until Section 3.3 and Section 5 (EXPERIMENTAL PROTOCOL). In the Abstract and Introduction, it is said that the barrier to success application of GANs to audio generation is the non-invertible spectral representation.
> >
> > If WaveGAN is what the paper introduces to overcome the non-invertible issue, it is confusing to see that SpecGAN outperforms WaveGAN by a large margin in Inception score. And I think, 58% accuracy for WaveGAN vs 66% for SpecGAN cannot be said to be similar. The non-invertible issue is not a issue.
> >
> > Phase shuffle increases Inception scores substantially (4.12->4.67) in WaveGAN, but deteriorate Inception score in SpecGAN. And there is no discussion about this.
> >
> > It is appreciated that the paper presents a nice effort to apply GANs to audio generation. But the presentation should be improved to make clearer the concrete methodological contributions and to present more consistent results.
> >
> > “““ Qualitative ratings are poor. ”””
> >
> > Around 60% accuracy for generated data is not a strong evidence that WaveGAN/SpecGAN as presented in this paper is promising. Thinking about generating digit images 0-9 by training GANs in MNIST. The labeling accuracy for generated images would be much higher than 60%. GAN based audio synthesis is interesting and should be promising. But the results shown in this paper does not fully validate this.
> >
> >
> > ===========  comments after reading response ===========
> >
> > The reviewer would like to thank the authors for their response, which clarifies some unclear issues. However, the response does not address my main concern about the algorithmic contribution of the proposed method.
> >
> > > While we outlined additional methodological contributions (e.g. phase shuffle) in response to your initial review, our *primary* contribution is still a GAN that operates on raw audio waveforms. Before this paper, the ability of GANs to generate one dimensional time series data had not been demonstrated.
> >
> > This seems to be a overstate. Pascual et al. (2017) (SEGAN) has already shown the ability of GANs to conditionally generate one dimensional time series data. Instead of simply saying that "Pascual et al. (2017) apply GANs to raw audio speech enhancement.", it would be better to provide more relevant comparisons, inform the readers that the difference between Pascual et al. (2017) and this paper is conditional generation vs unconditional generation, and clarifies the difficulty in unconditional generation.
> >
> > The paper consists of interesting efforts and contributions. I would like to suggest the authors to move the contributions of phase shuffle (Section 3.3) and the learned post processing filters (Appendix B) to the foreground. This presentation problem make me hold the scoring.

---

> > > ### Author Response · Authors · 2018-11-13
> > > **Author Response to Reviewer #3 (1/2)**
> > >
> > > Thank you for elaborating. We still do not understand what specifically caused you to *change* your initial score of 6 to a 5. Respectfully, these criticisms appear to be post-hoc justification for your updated score. Nevertheless, we will address your concerns below, and have updated the paper with minor revisions based on your feedback:
> > >
> > > ** Clarifying contributions **
> > >
> > > “““If as said in the response, the concrete methodological contributions are phase shuffle (Section 3.3) and the learned post processing filters (Appendix B). At first reading of the paper, these contributions are not clear. These statements are not presented until Section 3.3 and Section 5 (EXPERIMENTAL PROTOCOL). In the Abstract and Introduction, it is said that the barrier to success application of GANs to audio generation is the non-invertible spectral representation. ”””
> > >
> > > While we outlined additional methodological contributions (e.g. phase shuffle) in response to your initial review, our *primary* contribution is still a GAN that operates on raw audio waveforms. Before this paper, the ability of GANs to generate one dimensional time series data had not been demonstrated.
> > >
> > > ** WaveGAN vs SpecGAN **
> > >
> > > “““The non-invertible issue is not a issue.”””
> > >
> > > Simply put, the non-invertibility of SpecGAN *is* an issue. If you naively apply image GANs to audio generation (by operating on spectrograms i.e. SpecGAN), the non-invertibility of the spectrograms is a major barrier to downstream usability because the resultant audio quality is atrocious (see links below). While humans are able to label digits generated by SpecGAN with higher accuracy than those generated by WaveGAN, the human-assessed subjective sound quality of SpecGAN is worse (and a simple listening test confirms this).
> > >
> > > By operating directly on waveforms, our WaveGAN method achieves higher audio quality, is simpler to implement, and is a first for generative modeling of audio using GANs.
> > >
> > > WaveGAN (recognizable and better audio quality): http://iclr-wavegan-results.s3-website-us-east-1.amazonaws.com/wavegan_sc09.wav
> > > SpecGAN with *approximate* inversion (recognizable but poor audio quality): http://iclr-wavegan-results.s3-website-us-east-1.amazonaws.com/specgan_sc09.wav
> > >
> > > ** Presentation of qualitative ratings **
> > >
> > > “““Around 60% accuracy for generated data is not a strong evidence that WaveGAN/SpecGAN as presented in this paper is promising. Thinking about generating digit images 0-9 by training GANs in MNIST. The labeling accuracy for generated images would be much higher than 60%.”””
> > >
> > > It is unfair to compare our results to the hypothetical human labeling performance of digits generated by a GAN trained on MNIST. While MNIST may have the same number of semantics modes (10) as our SC09 digit dataset, these datasets are quite different in terms of dimensionality. Images in MNIST can be seen as vectors in 784-dimensional (28x28) space, whereas waveforms in SC09 are vectors in 16000-dimensional space. Higher dimensionality does not necessarily equate to greater difficulty to generative modeling, but it certainly should discourage direct comparison.
> > >
> > > We argue that our results are indeed “promising”. We developed and compared multiple methods for generating audio waveforms with GANs, a first for the field. Our results are analogous to early papers in image generation with GANs (e.g. DCGAN from ICLR 2016), and such results laid groundwork for remarkable breakthroughs in high-resolution image synthesis.
> > >
> > > ** Clarifying inception scores **
> > >
> > > “““If WaveGAN is what the paper introduces to overcome the non-invertible issue, it is confusing to see that SpecGAN outperforms WaveGAN by a large margin in Inception score. And I think, 58% accuracy for WaveGAN vs 66% for SpecGAN cannot be said to be similar. The non-invertible issue is not a issue.”””
> > >
> > > While many GAN papers use inception score as a primary evaluation metric, we state that our intention is to use human evaluations as our primary metric: “While inception score is a useful metric for hyperparameter validation, our ultimate goal is to produce examples that are intelligible to humans. To this end, we measure the ability of human annotators...”
> > >
> > > In our updated manuscript, we additionally hypothesize a reason behind the discrepancy between inception scores and subjective quality assessments: “this discrepancy [between inception score and human assessments of quality] can likely be attributed to the fact that inception scores are computed on spectrograms while subjective quality assessments are made by humans listening to waveforms.”
> > >
> > > Furthermore, while the focus of our paper is on WaveGAN as it is a non-trivial application of GANs to audio generation, we acknowledge that spectrogram-based methods also achieve reasonable results for our task despite audio quality issues: “We see promise in both waveform and spectrogram audio generation with GANs; our study does not suggest a decisive winner.”

---

> > > > ### Author Response · Authors · 2018-11-13
> > > > **Author Response to Reviewer #3 (2/2)**
> > > >
> > > > “““Phase shuffle increases Inception scores substantially (4.12->4.67) in WaveGAN, but deteriorate Inception score in SpecGAN. And there is no discussion about this.”””
> > > >
> > > > Applied to spectrograms, phase shuffle is a radically different operation than it is for waveform because spectrograms have more compact temporal axes, and we fill in jittered samples with padding. This means that, in the worst case, a SpecGAN discriminator with minimum phase shuffle (n=1) may be observing 468ms (nearly half the example) of padded waveform. On the other hand, a WaveGAN discriminator with n=1 observes a worst case of 83ms of padded waveform.
> > > >
> > > > We have added a sentence to our paper: “Phase shuffle decreased the inception score of SpecGAN, possibly because the operation has an exaggerated effect when applied to the compact temporal axis of spectrograms.”

---

### Author Response · Authors · 2018-11-08
**Author Response to Reviewers**

We would like to thank all of the reviewers for their thoughtful comments and suggestions. We have uploaded a new version of our manuscript with improvements based on reviewer feedback. Reviews were all positive for our paper (though one reviewer has since lowered their score without explanation), with reviewers highlighting the promising nature of our results as well as the clarity and reproducibility of our paper. We will respond to specific comments from each reviewer separately. If reviewers would like to provide additional context behind their scores we would be happy to provide feedback.

---

### Meta-Review · Area_Chair1 · 2018-12-16
**interesting application of GANs to audio, may spark further research.**

**Confidence:** 4
**Recommendation:** Accept (Poster)

**Metareview:**

This paper proposes a GAN model to synthesize raw-waveform audio by adapting the popular DC-GAN architecture to handle audio signals. Experimental results are reported on several datasets, including speech and instruments.

Unfortunately this paper received two low-quality reviews, with little signal. The only substantial review was mildly positive, highlighting the clarity, accessibility and reproducibility of the work, and expressing concerns about the relative lack of novelty. The AC shares this assessment. The paper claims to be the first successful GAN application operating directly on wave-forms. Whereas this is certainly an important contribution, it is less clear to the AC whether this contribution belongs to a venue such as ICLR, as opposed to ICASSP or Ismir.  This is a borderline paper, and the decision is ultimately relative to other submissions with similar scores. In this context, given the mainstream popularity of GANs for image modeling, the AC feels this paper can help spark significant further research in adversarial training for audio modeling, and therefore recommends acceptance. I also encourage the authors to address the issues raised by R1.